# Genome-Wide Analysis and Function Prediction of Long Noncoding RNAs in Sheep Pituitary Gland Associated with Sexual Maturation

**DOI:** 10.3390/genes11030320

**Published:** 2020-03-17

**Authors:** Hua Yang, Jianyu Ma, Zhibo Wang, Xiaolei Yao, Jie Zhao, Xinyue Zhao, Feng Wang, Yanli Zhang

**Affiliations:** Jiangsu Livestock Embryo Engineering Laboratory, Nanjing Agricultural University, Nanjing 210095, China; 15062275837@163.com (H.Y.); 2017105027@njau.edu.cn (J.M.); 2018105081@njau.edu.cn (Z.W.); 2016205004@njau.edu.cn (X.Y.); 2017105036@njau.edu.cn (J.Z.); 30216418@njau.edu.cn (X.Z.); caeet@njau.edu.cn (F.W.)

**Keywords:** noncoding RNA, pituitary, sheep, immature, mature

## Abstract

Long noncoding RNA (lncRNA) plays a crucial role in the hypothalamic-pituitary-testis (HPT) axis associated with sheep reproduction. The pituitary plays a connecting role in the HPT axis. However, little is known of their expression pattern and potential roles in the pituitary gland. To explore the potential lncRNAs that regulate the male sheep pituitary development and sexual maturation, we constructed immature and mature sheep pituitary cDNA libraries (three-month-old, TM, and nine-month-old, NM, respectively, n = 3) for lncRNA and mRNA high-throughput sequencing. Firstly, the expression of lncRNA and mRNA were comparatively analyzed. 2417 known lncRNAs and 1256 new lncRNAs were identified. Then, 193 differentially expressed (DE) lncRNAs and 1407 DE mRNAs were found in the pituitary between the two groups. Moreover, mRNA-lncRNA interaction network was constructed according to the target gene prediction of lncRNA and functional enrichment analysis. Five candidate lncRNAs and their targeted genes *HSD17B12*, *DCBLD2*, *PDPK1*, *GPX3* and *DLL1* that enriched in growth and reproduction related pathways were further filtered. Lastly, the interaction of candidate lncRNA TCONS_00066406 and its targeted gene *HSD17B12* were validated in in vitro of sheep pituitary cells. Our study provided a systematic presentation of lncRNAs and mRNAs in male sheep pituitary, which revealed the potential role of lncRNA in male reproduction.

## 1. Introduction

Long noncoding RNA (lncRNA), one class of noncoding RNA (ncRNA), with > 200 bp in length, no protein coding potential and poor conservatism has been studied in many biological processes including growth development [1], reproduction [2] and disease [3]. Many evidences have proved the function of lncRNA on transcription, post-transcription and protein translation levels by regulating the gene expression and chromatin remodeling [4]. There are a large number of action models of lncRNA have been discovered [5]. For instance, certain lncRNA can affect the expression of downstream gene by interacting with RNA polymerase II or chromatin; some lncRNAs can also affect the gene expression by means of reverse complementation among the sequence of genes, contributing to change of the protein activity and location through binding lncRNA that forms RNA-protein complex. In addition, some small ncRNAs including siRNA, piRNA and miRNA can be produced by processing lncRNA. Based on extensive research, numerous databases have been built [6] covering human, mouse and other specials, such as LNCipedia [7,8], lncRNAdb [9], NONCODE [10] and LncRNADisease [11].

The pituitary gland is an important endocrine gland, it plays a critical regulatory role in animal life activities by secreting various hormones, such as luteinizing hormone (LH), follicle-stimulating hormone (FSH) and growth hormone (GH) that effect on reproduction and growth, respectively [12]. In male animals, pituitary gonadotropins (LH and FSH) could maintain normal spermatogenic function through synergistically regulating the development of spermatogenic cells, Leydig cells and Sertoli cells in the testis and the synthesis and secretion of gonadal steroid hormones. Recently, a large amount of research has been identified the role of lncRNA in pituitary functions. For example, MIR205HG, a lncRNA that harbors the gene for miR-205, enabled to regulate the secretions of GH and prolactin in mouse anterior pituitary by modulating the Pit1 and Ztbt20 transcription factors but independently of miR-205 [13]. The study of lncRNA m433s1 function in rat anterior pituitary cells indicated that lncRNA m433s1 upregulated the expression of *FSHβ* and the FSH production through performing as a competing endogenous RNA (ceRNA) by sponging miR-433 which could regulate *FSHβ* expression [14,15]. In many tumor studies, lncRNA H19 is considered to be an oncogenic gene [16,17] and performed high expression in aggressive growth hormone-secreting pituitary adenomas (GHPAs). This indicated lncRNA H19 is a potential target for the study of invasive GHPAs [18]. LncRNA H19 could also be acted as a tumor suppressor to inhibit the pituitary tumor growth [19]. LncRNA SNHG1 activated the TGFBR2/SMAD3 and RAB11A/Wnt/β-catenin pathways to promote the progression of pituitary tumors by sponging miR-302/372/373/520 in pituitary tumor cells [20]. LncRNA RPSAP52 was also verified to promote cell growth by acting as sponge for miR-15a, miR-15b, and miR-16, which have been already described to be able to target *HMGA2* [21]. LncRNA CCAT2 was performed to interact with *PTTG1* to promote pituitary adenoma cell proliferation, migration, and invasion, which further had an influence on human pituitary adenoma development and progression [22]. In addition, the expression of lncRNA MEG3 and HOTAIR was also related to non-functioning pituitary adenomas (NFPAs) invasion [23,24]. LncRNA C5orf66-AS1 suppressed the development and invasion of pituitary null cell adenomas [25]. In pituitary prolactinoma, one of the most prevalent types in pituitary adenoma, lncRNA CLRN1-AS1 regulated cell growth and autophagy by acting as a ceRNA to suppress the Wnt/β-catenin signaling pathway [26]. In summary, many researches have been involved in pituitary diseases and focused on species such as humans, mice and rats.

Lots of studies have been explored in the research on lncRNA in male reproduction, female reproduction and growth development of sheep, including testis [27], ovary [28], uterus [29], skeletal muscle [28] and adipose tissues [30]. For the functional research of pituitary in sheep reproduction, there were 235 DE lncRNAs have been identified in embryonic and adult female sheep pituitary. Several lncRNAs such as LNC_001056, LNC_000322 and LNC_000207 were associated with hormone secretion and pituitary gland development [31]. Fifty-seven DE lncRNAs and 298 DE mRNAs were found between high prolificacy and low prolificacy female sheep pituitary, the potential role of candidate lncRNA MSTRG.259847.2 and its target gene *SMAD2* were investigated in sheep pituitary cells in vitro [32]. Moreover, the sexual maturity period is a key point for male animals, and the differential expression pattern of lncRNAs in immature and mature rat pituitary have been detected and there are 1181 significantly DE lncRNAs transcripts that were identified [33]. However, there is no study about the potential role of lncRNAs involved in male sheep pituitary, especially in immature and mature sheep pituitary. Meanwhile, Hu sheep is a famous high-fertility sheep breed in China, and Hu sheep enters sexual maturity after about six-months-old. Therefore, it is meaningful to explore the function of the pituitary in male Hu sheep reproduction and growth development during sexual maturity.

To identify the differential expression profiles of lncRNAs and mRNAs in immature and mature sheep pituitary, we constructed six cDNA libraries including three 3-month-old (TM) and three 9-month-old (NM) sheep pituitary tissues. In addition, based on bioinformatics analysis, several candidate DE lncRNAs and their targeted genes were screened which potentially play roles on sheep growth and reproduction, and the interaction of lncRNA TCONS_00066406 and targeted gene *HSD17B12* was also validated in in vitro sheep pituitary cells. Our study firstly revealed the comprehensive analysis of lncRNA and mRNA in sheep pituitary, further providing the candidate lncRNAs with potential functions in growth development and reproduction.

## 2. Materials and Methods

All the experimental methods were conducted in accordance with the approval of the Animal Experiments of Nanjing Agricultural University, China, and the animal breeding experiment were approved by the Animal Care and Use Committee of Nanjing Agricultural University, China (Approval ID: SYXK2011–0036).

### 2.1. Animals and Samples Collection

Six healthy male Hu sheep with similar body condition of three-months-old and nine-months-old (TM and NM, n = 3) were fed in a same condition, including diet, drinking and feeding environment at the Jiangsu Hailun Sheep Industry Limited Company, Jiangyan, China, and were selected and slaughtered for pituitary collection. After slaughtering, we manually sawed the head with a chainsaw, then the pituitaries were removed and collected with RNAlater and snap frozen in liquid nitrogen immediately for RNA extraction. In addition, the testis tissues of five-day-old (FD), TM, six-month-old (SM), NM and two-year-old (TY) sheep were collected for histomorphological analysis using hematoxylin-eosin staining. As described in our previous study [34], we found that the cell types in the tubules gradually increased with age. It is obvious to find that there are sperm cells in NM testis compared with TM and SM. Therefore, to ensure that sheep reach sexual maturity, we chose NM as the sexual maturity stage.

### 2.2. Plasma Hormone Assay

Before slaughtering, the blood samples were collected via jugular vein and centrifuged at 3000 rpm for 10 min, the supernatant of blood was obtained and stored at −20 °C for hormone level analysis by ELISA following the instructions of GH, FSH and LH ELISA kits purchased from Kmaels Biotech Co., Ltd.

### 2.3. RNA Extraction and Strand-Specific cDNA Library Construction

The total RNA of six pituitary samples was extracted using TRIzol (Invitrogen Life Technologies, Carlsbad, CA, USA) and the RNA purity, concentration and quality were detected by NanoDrop (NanoDrop Technologies, Wilmington, DE, USA), Aglient 2100 and electrophoresis, respectively. Then ribosomal RNAs (rRNAs) were removed to retain mRNAs and ncRNAs using epicenter Ribo-Zero^TM^ regent. The remain RNAs were randomly fragmentated into short fragments which were used for cDNA synthesis. We performed the library construction use the method of strand-specific according to the following steps. First-strand cDNA was synthesized using random hexamers with rRNA-depleted RNA as a template. dNTPs, RNase H and DNA polymerase I were used for second-strand synthesis. And the cDNA was purified using AMPure XP beads. Next, cDNA fragments purification, end repair, poly(A) addition and Illumina sequencing adapters ligation were carried out with the QiaQuick PCR extraction kit. Then UNG (Uracil-N-Glycosylase) was used to digest the second-strand cDNA. cDNA fragments were size selected by agarose gel electrophoresis, PCR amplified, and sequenced using Illumina HiSeqTM 4000 by Gene Denovo Biotechnology Co. (Guangzhou, China).

### 2.4. Data Analysis and Transcripts Assembly

To ensure the data quality, the reads containing adapters, consisting of all A bases, containing more than 10% of unknown nucleotides (N) or containing more than 50% of low quality (Q-value ≤ 20) bases were filtered. Then, the clean reads of each sample were then mapped to reference genome Oar_ v1.0 by TopHat2 [35] (version 2.1.1, Johns Hopkins University, Baltimore, MD, USA), respectively. Then the transcripts were assembled using Cufflinks software (version 2.2.1, University of Washington, Seattle, WA, USA) [36], and the transcripts with ≥200 bp in length and the number of exon ≥ 2 were filtered to identify whether it is mRNA or lncRNA. After aligned with the reference genome, the known transcripts were identified and the unmapped reads were used for reconstructing transcripts and to obtain novel transcripts.

### 2.5. New lncRNA Identification and Classification

Firstly, we performed the transcripts reconstructing (identified new transcripts) by Cufflinks software, and then two software CNCI (version 2014, Chinese Academy of Sciences, Beijing, China) [37] and CPC [38] were used to assess the protein-coding potential of novel transcripts. The CPC and CNCI score < 0 was defined as noncoding. SwissProt database was used for detecting the protein annotation information of new transcripts. Then, the novel transcripts with non-protein-coding potential and non-protein annotation were identified as novel lncRNAs. LncRNAs were classified into five types based on the position of lncRNA relative to protein-coding genes in the genome: intergenic lncRNAs, bidirectional lncRNAs, intronic lncRNAs, antisense lncRNAs, and sense overlapping lncRNAs, as different types of lncRNAs may play different biological functions.

### 2.6. Differential Expression Analysis of mRNA and lncRNA

The DE transcripts of mRNAs and lncRNAs were analyzed respectively. The dDE analysis was performed using edgeR package, and the *p*-value < 0.05 and fold change > 2 were the filtering criteria to identify significant differential expressed genes (DEGs). Then the differential expression pattern clustering was drawn by Gene Denovo platform.

### 2.7. Functional Enrichment Analysis of DEGs

Gene Ontology (GO) [39] including biological process (BP), cellular component (CC) and molecular function (MF) provides all GO terms that significantly enriched in DEGs comparing to the genome background, and filter the DEGs that correspond to biological functions. The Kyoto Encyclopedia of Genes and Genomes (KEGG) pathway-related database [40] was also used to identify significantly enriched metabolic pathways or signal transduction pathways in DEGs comparing with the whole genome background. To recognize the main biological functions that DEGs exercise, all DEGs were mapped to GO terms or KEGG pathways in GO and KEGG databases, and the degree of enrichment was calculated by *p*-value that went through FDR correction; the GO term or pathway with FDR ≤ 0.05 was defined as significantly enriched pathways in DEGs.

### 2.8. Targeted Gene Prediction Analysis of lncRNA

To further explore the importance of lncRNA in pituitary functions, we predicted the targeted genes of lncRNAs according to antisense, cis-act and trans-act analysis. First, antisense lncRNA analysis refers to the antisense lncRNA binding to the sense strand of mRNA to regulate the post-transcriptional processes. Second, lncRNA cis-regulated analysis means lncRNAs are cis-regulating with their neighboring genes within 10 kb on the same allele in transcriptional or post-transcriptional level. LncRNA trans-regulation Analysis is based on the correlation coefficient between lncRNA and mRNA, and the value > 0.9 is considered to be trans-action. Based on these three interact relationship, pituitary function related genes and its target-interacted lncRNAs were filtered.

### 2.9. lncRNA-mRNA Co-Expressed Network Construction

The interaction network between pituitary function related genes and potential lncRNAs was constructed by Cytoscape software. The network between PPI (protein–protein interactions) candidate genes that screened by STRING database was also constructed by Cytoscape software.

### 2.10. Sheep Pituitary Cell Isolation and Cell Transfection

To verify the interaction relationship of candidate DE lncRNAs and its targeted genes, sheep pituitary cells were isolated and identified using two-step enzymatic digestion and immunofluorescence staining of FSH (Bioss, bs-1536R; Beijing, China,), respectively based on the previous research [32]. The siRNA of lncRNA TCONS_00066406 was synthesized by the GenePharma company, Shanghai, China. The sequences of siRNA are listed in Appendix A. Moreover, RNA interference was performed follow Lipofectamine 3000 reagent protocols (Invitrogen Life Technologies, Carlsbad, CA, USA), then the cells were harvested for qPCR after incubation for 48 h.

### 2.11. Real-time PCR Analysis

Real-time PCR (qPCR) was performed to detect the expression of DE genes and lncRNAs by QuantStudio™ Real-Time PCR system (Life Technologies, Carlsbad, CA, USA) with SYBR Green Master mix (Roche Applied Science, Mannheim, Germany). The reaction system contained 10 µL SYBR, 7.8 µL RNase-free water, 0.6 µL forward primer, 0.6 µL reverse primer and 1 µL cDNA. The PCR reaction system was followed by the regent of SYBR Green Master mix as previous description. The primer sequences were presented in Appendix A.

### 2.12. Statistical Analysis

All data were expressed as mean ± the standard errors of the means and analyzed using the SPSS software (version 20.0, IBM company, Armonk, NY, USA). The multigroup comparisons of the means were analyzed using one-way analysis of variance (ANOVA). Posthoc contrasts were implemented using the Tukey test. *p*-values of <0.05 were considered statistically significant. Each group contained three biological repetitions and three technical repetitions per data.

## 3. Results

### 3.1. The Hormone Levels in Immature and Mature Sheep

To investigate the function of pituitary in reproduction and growth development, the hormone levels of GH, LH and FSH in immature and mature sheep blood were detected by ELISA. It showed that the levels of these three hormones were extremely significant higher in the NM group than in the TM group (*p* < 0.01, Figure 1).

### 3.2. Overview of Sequencing in Sheep Pituitary Tissues

In our study, we identified lncRNA and mRNA through constructing six cDNA libraries, denoted TM1, TM2, TM3 and NM1, NM2, NM3, and each cDNA library obtained an average of 10.65 Gb clean data after sequencing. After further filtering, clean data of high quality were obtained, and the percentage of bases with the quality of reaching Q30 (also representing that the sequencing accuracy is 99.9%) was more than 95.43% averagely. The GC content of each library was more than 43.25%. In addition, more than 92.49% of clean reads could be mapped to the reference genome and the unique mapping ratio of reads was more than 90.67% (Appendix A).

### 3.3. Identification of lncRNAs and mRNAs in Sheep Pituitary

After mapping reads to the reference genome, the transcripts were assembled using Cufflinks, then the known mRNAs and lncRNAs were identified. Based on the location of transcripts in genome, the transcripts with the length ≥ 200 bp and the number of exon ≥ 2 were filtered for new mRNA and lncRNA identification. For new lncRNAs prediction, CPC and CNCI were used for potential coding ability detection, and SwissProt, a protein database, was used for protein annotation information analysis (Figure 2a). The transcripts with non-coding ability and non-protein annotation were considered to be new lncRNAs. In our study, 2417 known lncRNAs, 33,746 known mRNAs, 1256 new lncRNAs and 8575 new mRNAs were identified respectively. According to the position of the new lncRNA on the genome relative to the protein-encoding gene, the new lncRNAs could be divided into five categories, the largest proportion of which was intergenic lncRNAs and there was no intronic lncRNAs identified (Figure 2b). Compared to mRNA with 12.53 exons, lncRNA had 3.27 exons on average and most of lncRNAs had 2–3 exons (Figure 3a). All lncRNAs and mRNAs had similar chromosome distribution, and there was a large proportion on chromosome 1, 2 and 3. Moreover, there were no lncRNA distributed in mitochondrial chromosome (Figure 3b). The average length of lncRNAs and mRNAs was 24,823.95 bp and 68,495.69 bp, respectively (Figure 3c). The expression level of lncRNAs was lower than mRNAs regardless of being in the TM or in the NM group (Figure 3d). In our study, the expression information of lncRNAs and mRNAs in sheep pituitary was systematically integrated.

### 3.4. Differentially Expressed lncRNAs and mRNAs in Immature and Mature Sheep Pituitary

To screen the DE lncRNAs and mRNAs between immature and mature sheep pituitary, the criteria with fold change > 2 and *p* < 0.05 was performed to filter DE lncRNAs and mRNAs. In our study, there were 193 DE lncRNAs included, of which 68 upregulated and 125 downregulated, and 1407 DE mRNAs, including 640 upregulated and 767 downregulated in the NM group compared with the TM group (Figure 4). In order to verify the sequencing accuracy, 10 DE transcripts were randomly selected to validate their relative expression levels in pituitary of TM and NM groups using qPCR. The results showed that the expression changes were consistent with RNA-seq results, which indicating the reliability of sequencing data (Figure 5).

### 3.5. Functional Enrichment Analysis and Crucial Gene Filter of DE mRNAs

GO and KEGG enrichment analysis were used to analyze the function of DEGs. In our study, 521 DEGs can be mapped to GO database, and the top 15 significantly enriched GO terms were listed in Figure 5a. In the BP part, the genes were significantly enriched in response to hormone, chromatin related function, apoptotic and corticosteroid receptor signaling pathways. In CC part, the genes were mainly located in intracellular part and organelle. The DEGs were also related to compound binding and enzyme activity. Afterwards, 503 DEGs that can be annotated by KEGG database. Based on top 20 of pathway enrichment analysis, it is indicated that Lysine degradation singling pathways was the most significant enrichment pathway. In addition, many DEGs were also significantly enriched in Rap1, mTOR, cAMP, FoxO and Hippo signaling pathways (Figure 6b). To investigate the function of pituitary in growth and reproduction, 163 DEGs (77 up-regulated and 86 down-regulated) that enriched in growth, reproduction and steroid hormone related GO terms were selected. As shown in Appendix A, several terms such as hormone related terms, growth and tissue development were significantly enriched (*p* < 0.05). Additionally, 156 DEGs (76 up-regulated and 80 down-regulated) were enriched in growth, reproduction and steroid hormone related pathways (Figure 7a). Above all, these 287 DEGs were filtered to predict interaction relationship and construct interaction network. After interrelationship analysis of these DEGs, 98 mRNAs with correlation coefficient > 0.999 and *p* < 0.05 were selected to construct interaction network. (Figure 7b).

### 3.6. Interaction Analysis of Candidate lncRNAs and Their Targeted mRNAs

To explore the roles of lncRNAs on pituitary function, the predicted antisense-, cis- and trans- targeted mRNAs of DE lncRNAs were performed functional enrichment analysis. As shown in Appendix A, we constructed the interaction network between DE lncRNAs and its antisense- and cis- targeted mRNAs. LncRNA XR_00102479.3 and XR_003585973.1 can interact with eight and five mRNA transcripts of *ZNF705A* and *SCAP*, respectively, and different mRNA transcripts were differently expressed in immature and mature pituitary, which indicated the potential effect of lncRNA in mRNA level. Moreover, according to the enrichment analysis of targeted genes, we filtered lncRNA TCONS_00066406, XR_003591760.1, TCONS_00084471, TCONS_00032215, TCONS_00045021 and their five targeted genes *HSD17B12*, *DCBLD2*, *PDPK1*, *GPX3*, *DLL1* in Table 1, containing two cis-, two antisense- and one trans- interaction relationships.

### 3.7. Verification of Targeting Relationship between lncRNA TCONS_00066406 and Its Targeted Gene HSD17B12 in Sheep Pituitary

To identify the functional role of candidate lncRNAs in sheep pituitary, we detected the expression level of lncRNA TCONS_00066406 and its targeted gene *HSD17B12* in sheep visceral, intestinal tract and reproductive axis. The expression levels of lncRNA TCONS_00066406 and its targeted gene *HSD17B12* were higher in testis, pituitary and hypothalamus tissues than other organs (Figure 8a,b). Based on previous methods [28], we isolated sheep pituitary cells, and immunofluorescence analysis showed the marker gene *FSH* expressed in most cells (Figure 8c) and the gel electrophoresis results of PCR products indicated *FSH* and *LH* were expressed in the isolated cells (Figure 8d), which verified the isolated cells were pituitary cells. After RNA interference of lncRNA TCONS_00066406, the expression level of *HSD17B12* and *FSH*, *LH* were decreased (Figure 8e, *p* < 0.05).

## 4. Discussion

With the development of high-throughput sequencing technology, the sheep genome has been assembled and many noo-coding RNAs (ncRNAs) were annotated. LncRNA and other small ncRNAs, such as miRNA, piRNA and siRNA, have been explored the effect on growth development, reproduction, metabolism and diseases research via participation in regulating transcription levels, post-transcription levels and protein levels. Some research has investigated the role of lncRNA on growth development in muscles [41], heart [42], brain [43], spinal cord neuron [44] and cancer cells [45]. In addition, hypothalamic-pituitary-gonadal (HPG) axis plays an important role in reproduction, and there are many studies focus on hypothalamus [46,47], testis [48] and ovary [49] during different development stages. The pituitary gland is essential for mammalian reproduction and growth development owing to its function of GH, FSH, LH, and PRL hormone secretion. Micro RNA [50], circular RNA [51] and lncRNA [31] were performed in different stages of sheep pituitary, however, little research of lncRNA on immature and mature stage of male sheep pituitary have been explored. In our study, three-month-old and nine-month-old male sheep were considered to be immature and mature stages for further lncRNA identification in the pituitary, and our study provided the systematic description and differential expression profiles of lncRNA and mRNA in sheep pituitary.

Hormone level is a crucial factor in individual development, and GH, FSH and LH are mainly secreted via the pituitary, regulating growth and reproduction, respectively. In our study, we found the hormone levels were significantly increased with age which confirmed the promotion of growth and reproduction capacity after sheep sexual maturity. There are studies investigating the regulation of lncRNAs on GH [13], FSH [14,15] and LH [32] secretions, to verify whether lncRNA regulates the hormone synthesis in the pituitary during male sheep sexual maturity, and find candidate lncRNA with potential functions in reproduction and growth development. The expression profiles of lncRNAs and mRNAs were described. In our study, we determined 3673 lncRNAs, compared with 19,672 lncRNAs [32] and 1755 lncRNAs [31] which were identified in other sheep pituitary research, which possibly attributed to the different reference genome versions including Oar_ v1.0, Oar_ v4.0 and Oar_ v3,1. Different lncRNA prediction software were used for sequencing. In addition, the number of exon contained in lncRNA was mainly 2–3 and the expression level of lncRNA was lower than mRNA, that was similar with the female sheep pituitary [31]. Consistent with previous research [32,33], intergenic lncRNAs was the most abundant class of lncRNAs. Owing to less lncRNA identification in sheep than that in rat (3673 vs. 7039 lncRNAs), the number of DE lncRNAs was 193 in our study compared to 1181 DE lncRNAs filtered in rat immature and mature pituitary [33].

The GO and KEGG enrichment analysis were performed to annotate DEGs, in our study, the top 20 enriched pathways such as cAMP, Hippo, mTOR and FoxO were significantly enriched. Several studies have demonstrated that cAMP signaling plays a crucial role in the pituitary gland, regulating cellular growth and proliferation, hormone production and release [52,53] and the deregulation of the cAMP signaling pathway is a common occurrence in pituitary tumorigenesis [54]. The Hippo pathway regulates organ growth through the control of stem cell activity, proliferation and apoptosis. It has been reported that Hippo pathway play roles on pituitary development and regulation of pituitary uncommitted stem/progenitor cells [55]. A previous study documented that the mTOR singling pathway effected the secretion of growth hormones by interacting with the PI3K and Akt pathways [56]. In addition, it has been suggested that overexpression of FoxO1 in pituitary decreased the expression of *FSHβ* [57]. Therefore, these significantly enriched pathways can indirectly show that the DEGs play roles in pituitary function. To further filter the genes that play roles on pituitary functions including reproduction and growth, the gene interaction network was constructed and several filtered genes related to hormone secretions of pituitary gland were selected. For example, the expression of up-regulated gene *GH1* was consistent with the hormone level of GH in immature and mature sheep, which also can interact with many DE genes which indicated the importance of pituitary function on sheep growth development. The previous study proved the expression of *CTSB* was affected by the LH hormone level, and higher LH concentration could reduce *CTSB* expression [28]. In our study, the *CTSB* expression level was lower in mature sheep, on the contrary, LH hormone level increased in mature sheep. The down-regulated gene *LATS1* was filtered and it has been also confirmed to be related to LH secretion, *LATS1*-\-mice displayed LH deficiency and hyperplastic changes of pituitary and further effected the reproductive capacity [58]. *SALL1* can be co-located in GH, LH and FSH secretion cells which indicated the potential role of *SALL1* on pituitary hormone synthesis [59]. Up-regulated gene *CLOCK* [60], *NUMB* [61] and down-regulated gene *JAK1* [62] were found to be involved in pituitary development, *CLOCK* played a key regulatory role in gonadotropin release of pituitary, as well [63]. Down-regulated gene *DDR1* was investigated that promoting pituitary adenoma cell proliferation and invasion by reason of increasing expression of *DDR1* in hypoxia condition [64]. Sox family of transcription factors can regulate the embryonic development and the determination of cell fate, such as *SOX2* and *SOX3* were pivotal in pituitary function exertion [12]. *SOX5*, up-regulated in our study, was also upregulated in invasive pituitary tumor tissues and considered as an oncogene [65]. The fact that all these genes in the interaction network were enriched in reproduction and growth development related pathways provided the foundation for the study of pituitary function.

For filtering potential lncRNAs involved in pituitary function, antisense, cis-act and trans-act analysis were used for lncRNA targeted genes prediction and functional enrichment analysis was performed for targeted genes annotation. To strictly screen lncRNA and mRNA with targeting relationship, we concentrated on cis- and antisense- interaction, and mainly filter potential lncRNAs based on these two relationships. In our study, *HSD17B12* was enriched for steroid hormone biosynthesis and the expression level of *HSD17B12*, *FSH* and *LH* were decreased after disturbing the expression of lncRNA TCONS_00066406. A previous study has shown that *HSD17B12* can effect female reproduction through regulating prostaglandin synthesis [66], which further proved the importance of *HSD17B12* in reproduction and prompted the potential role of antisense-interact lncRNA TCONS_00066406 on reproduction. *DCBLD2* was recognized as the crucial regulation factor in tumorigenesis [67] and we found it was enriched in growth related pathways, suggesting lncRNA XR_003591760.1 possibly regulates pituitary development by interacting with *DCBLD2.* The targeted gene *PDPK1* of lncRNA TCONS_00084471 was enriched in several reproduction-related pathways indicated the potential influence of lncRNA TCONS_00084471 involved in reproduction regulating by pituitary. It also makes sense to study the candidate gene *GPX3* and *DLL1* and their targeted cis-interacted lncRNAs TCONS_00032215 and TCONS_00045021, respectively. In addition, we also predicted lncRNAs interacting with *GHR*, *GNRHR*, *IGF1* and *LHCGR* which have been reported to be related to growth and reproduction [68,69,70]. Five up-regulated lncRNAs XR_003587320.1, XR_003588821.1, XR_003587321.1, TCONS_00068108 and TCONS_00056664 were trans-interacted with *GHR*, *GNRHR*, *IGF1* and *LHCGR*, which indicated the potential role of these lncRNAs in growth and reproduction. However, the interacted lncRNAs of *GH*, *FSH* and *LH* were not predicted, and for investigating the regulation mechanism of candidate lncRNAs, we should seek further verification in vitro.

In our study, we firstly explored the differential expression of lncRNAs and mRNAs in immature and mature male sheep pituitary, and at the same time several lncRNAs and targeted genes were identified to be related with growth and reproduction, which were considered as the roles of lncRNAs in pituitary functions. Our results also provided the crucial information for the study of pituitary function in transcription level and supplied new ideas for the regulation of growth and reproduction in the secretion level of the pituitary.

## 5. Conclusions

In summary, we identified 193 and 1407 DE lncRNAs and mRNAs between immature and mature sheep pituitaries respectively using RNA sequencing. Additionally, an lncRNA–mRNA interaction network was constructed and several candidate lncRNAs which potentially related to reproduction and growth development were screened based on bioinformatics analysis. In addition, the targeting relationship of candidate lncRNA TCONS_00066406 and its targeted gene *HSD17B12* was verified in in vitro sheep pituitary cells.

The present research firstly provided comprehensive analysis of lncRNA and mRNA in sheep pituitary during male sexual maturity, and revealed the potential functions of lncRNAs in sheep growth development and reproduction.

## Figures and Tables

**Figure 1 genes-11-00320-f001:**
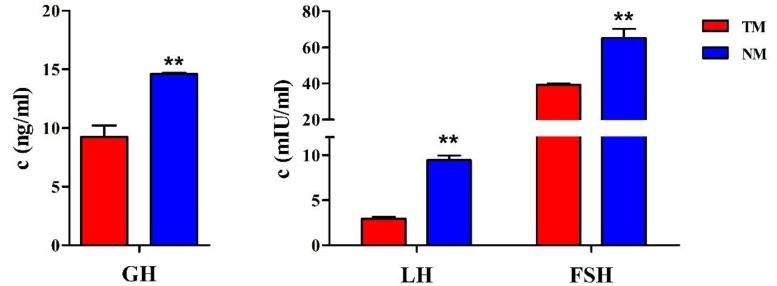
The hormone levels in immature and mature sheep blood. The red and blue boxes represent immature (3-month-old, TM) and mature (9-month-old, NM) sheep, respectively. The results are expressed relative to the TM group as the mean values ± SEM. ∗∗ denotes the extremely significant difference between two groups, *p* < 0.01.

**Figure 2 genes-11-00320-f002:**
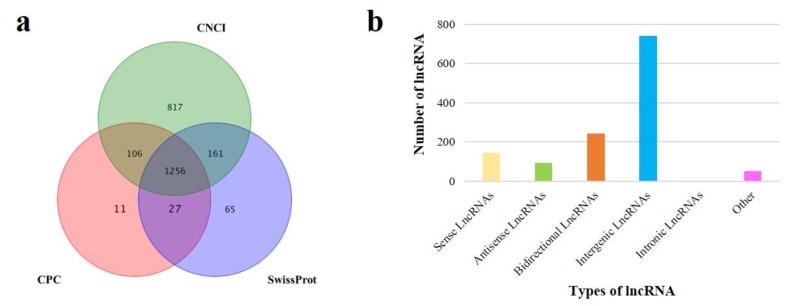
New lncRNA identification and classification. (**a**) The novel lncRNAs were identified by three predicted software CNCI, CPC and SwissProt. (**b**) The number statistics of different types of new lncRNAs.

**Figure 3 genes-11-00320-f003:**
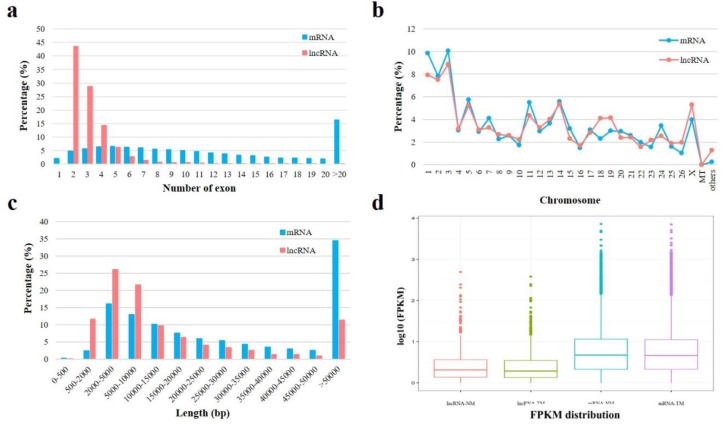
Comparison analysis of lncRNAs and mRNAs in sheep pituitary. (**a**) The statistics on the number of exon in lncRNAs and mRNAs. (**b**) The chromosome distribution of lncRNAs and mRNAs. (**c**) The length comparison of lncRNAs and mRNAs. (**d**) The boxplot shows the average expression levels (log10 (FPKM)) of lncRNAs and mRNAs in TM and NM groups.

**Figure 4 genes-11-00320-f004:**
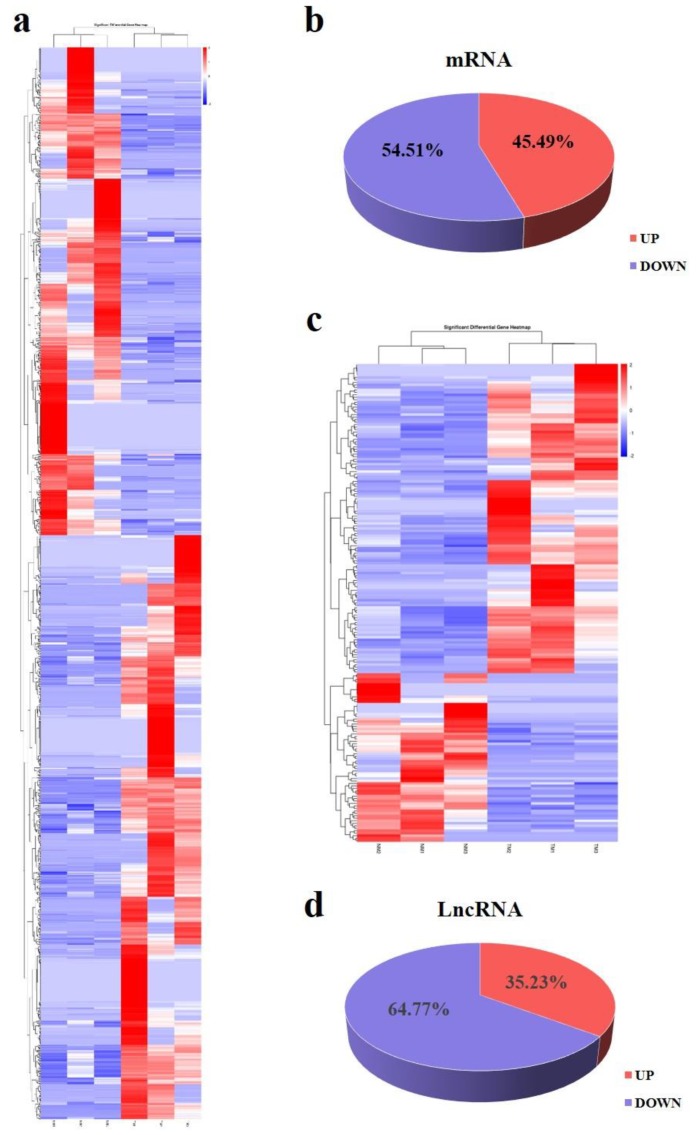
The differential expressed (DE) lncRNAs and mRNAs analysis: (**a**) and (**c**) represent the volcano plot of DE mRNAs and lncRNAs, respectively; (**b**) and (**d**) represent the up- and down-regulated ratio of DE mRNAs and lncRNAs, respectively. The red to blue color represents the fold change value from high to low.

**Figure 5 genes-11-00320-f005:**
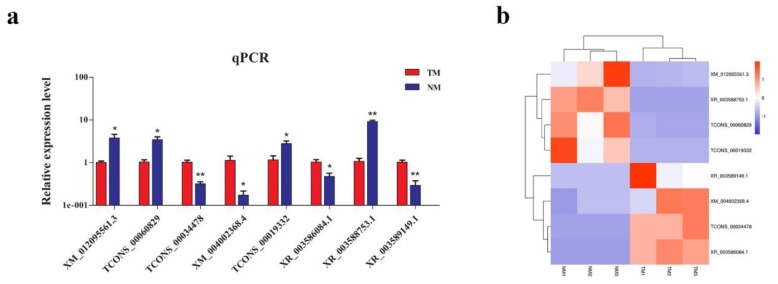
The verification of DE lncRNAs and mRNAs by qPCR. (**a**) Eight lncRNAs and mRNAs were randomly selected for qPCR verification and (**b**) shows the RNA-seq results of these eight DE lncRNAs and mRNAs. The qPCR results were expressed relative to the TM group as the mean values ± SEM and the relative expression levels were normalized to the expression levels of *GAPDH*. ** and * denotes the extremely significant and significant difference between two groups, *p* < 0.01 and *p* < 0.05.

**Figure 6 genes-11-00320-f006:**
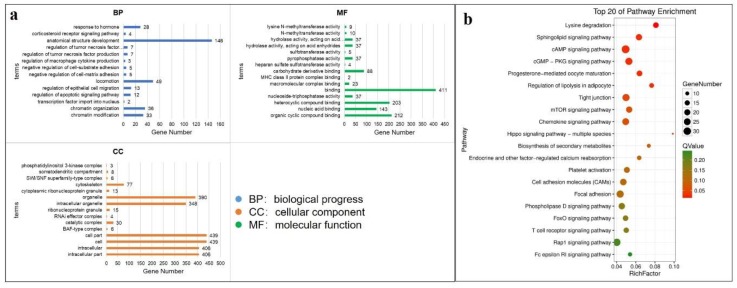
The top gene ontology (GO) and KEGG enrichment analysis. (**a**) The top 15 GO enrichment analysis of differential expressed genes (DEGs). (**b**) The top 20 KEGG enrichment analysis of DEGs.

**Figure 7 genes-11-00320-f007:**
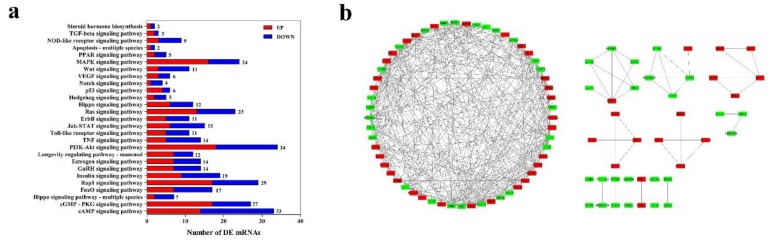
The crucial DE mRNAs screen. (**a**) The statistics of reproduction and growth related KEGG pathways of filtered DE mRNAs. (**b**) The network construction of 98 filtered DE mRNAs. Green and red represent the down- and up-regulated mRNAs, respectively.

**Figure 8 genes-11-00320-f008:**
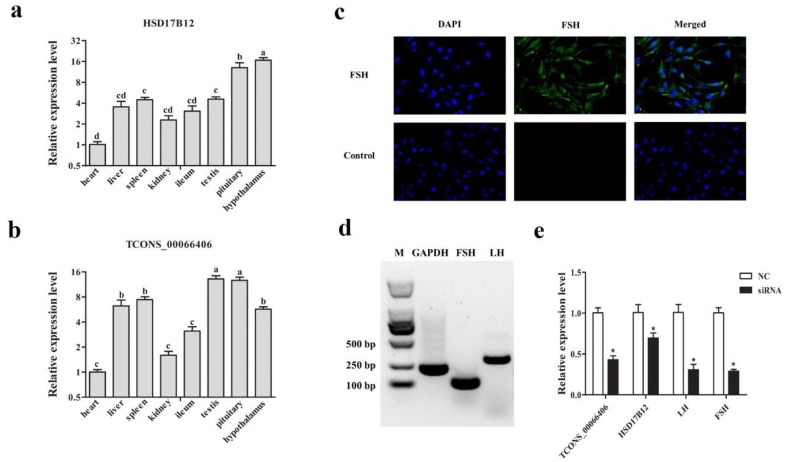
The validation of the interaction between lncRNA TCONS_00066406 and its targeted gene *HSD17B12*, with (**a**) and (**b**) representing the expression levels of *HSD17B12* and lncRNA TCONS_00066406 in different tissues, respectively. (**c**) The immunofluorescence staining of FSH in isolated cells. (**d**) The gel electrophoresis results of *FSH* and *LH* PCR amplification products. (**e**) The relative expression of lncRNA TCONS_00066406, HSD17B12, *FSH* and *LH* after sheep pituitary cell transfection using TCONS_00066406 siRNA. The qPCR results were expressed relative to the TM group as the mean values ± SEM and the relative expression levels were normalized to the expression levels of *GAPDH*. a, b and c: different letters denote statistically significant differences within each group; * denotes the significant difference between two groups, *p* < 0.05.

**Table 1 genes-11-00320-t001:** Five candidate long noncoding RNAs (lncRNAs) and their targeted genes.

LncRNA	UP/DOWN	Targeted Gene	Relationship	Mainly Pathways Enrichment
TCONS_00066406	UP	*HSD17B12*	antisense	Steroid hormone biosynthesis
XR_003591760.1	DOWN	*DCBLD2*	antisense	regulation of growth; regulation of cell growth
TCONS_00084471	DOWN	*PDPK1*	cis, upstream	PI3K-Akt signaling pathway; mTOR signaling pathway; AMPK signaling pathway; Apoptosis; FoxO signaling pathway; Thyroid hormone signaling pathway; PPAR signaling pathway
TCONS_00032215	DOWN	*GPX3*	cis, downstream	Thyroid hormone synthesis; cellular process
TCONS_00045021	UP	*DLL1*	cis, upstream	Notch signaling pathway

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
