# Peer review of "Genome-Wide Analysis and Function Prediction of Long Noncoding RNAs in Sheep Pituitary Gland Associated with Sexual Maturation"

_genes, 2020, doi:10.3390/genes11030320_

Round 1

Reviewer 1 Report

In this manuscript, Yang et al., dissect and sequence transcriptome data from pituitary of immature (3 month old) and sexually mature (9 month old) sheep. This manuscript offers an incremental improvement in our understanding of gene expression in the pituitary gland of sheep.

Major Revisions:

The authors relate differential gene expression within the pituitary gland to sexual maturity without the apparent acknowledgement the genes may be differentially related simply do to age, diet, etc.

Of course, mammals achieve sexual maturity with age but the authors are simply using age as a stand-in for sexual maturity of the testis. It would more suitable to have done semen analysis or identified similar age individuals with variable fertility.

If the authors identified 9 month old sheep with high/low fertility or 9 month old sheep, some having reached sexual maturity and others still lacking, the study would be more complete. The authors conclusion suffer from confirmation bias. They identified pathways associated with growth and reproduction and then fit the DE genes to those pathways without mention of other pathways showing enrichment of DE genes. The Discussion section primarily restates the results of the study with little to no interpretation of the data.

As such, the manuscript comes across as a "data" paper without offering any mechanism.

Minor Revisions:

There must be editing and correction of language and word usage. Soundness of experiments and scientific rigor is difficult to determine sometimes as the vocabulary and language has been, perhaps, misinterpreted. Section 2.6 in Materials and Methods: The authors utilize criteria such as p<0.05 and fold change>2 but neglect to include the false discovery rate as is the more accepted metric when using EdgeR for DE gene analysis.

The p-value may or may not be relevant but using FDR controls for multiple testing error. It doesn't appear the cDNA libraries were constructed in a strand-specific manner however the authors make assumptions about strandedness of the transcripts and use this to identify antisense lncRNA. These assumptions are likely based solely on canonical splicing but the authors do not state this.

Further, current assemblies and gene predictions result in many possible non-canonical splice sites. I cannot determine if the sequencing data has been submitted to public repositories or not. Sections 3.3: Authors claim to identify 1,256 new lncRNA and 8,575 new mRNA. This seems extraordinary.

The authors do not propose a reason for identifying so many novel transcripts. Are these simply new isoforms of currently annotated genes? Many of the novel lncRNA are intergenic. Are these relatively close to existing coding gene annotations? If so, it is entirely possible the novel transcripts are novel UTR exons in protein-coding genes that simply were not previously annotated. 

Author Response

Response to Reviewer 1 Comments

In this manuscript, Yang et al., dissect and sequence transcriptome data from pituitary of immature (3 month old) and sexually mature (9 month old) sheep. This manuscript offers an incremental improvement in our understanding of gene expression in the pituitary gland of sheep.

Major Revisions:

Point 1: The authors relate differential gene expression within the pituitary gland to sexual maturity without the apparent acknowledgement the genes may be differentially related simply do to age, diet, etc.

Response 1: Thank you so much for your comments. In our study, we explored the differential expression of mRNAs and lncRNAs between immature and mature sheep pituitary tissues (n=3). The six sheep were fed in a same condition, including diet, drinking and living environment. Furthermore, based on the production practice, 3-month-old (3M) and 9-month-old (9M) were identified the immature and mature stage, respectively. And we have added this information in line 105-106.

Point 2: Of course, mammals achieve sexual maturity with age but the authors are simply using age as a stand-in for sexual maturity of the testis. It would more suitable to have done semen analysis or identified similar age individuals with variable fertility.

Response 2: Thanks for your comments. We identified 3-month-old and 9-month-old as immature and mature stage that was same as our previous study [1]. And we have detected the testis tissue sections of 5-day-old (D5), 3M, 6-month-old (6M), 9M, and 2-year-old (2Y), and we could found that the cell types in the tubules gradually increased with age. It is obvious to find that there are sperm cells in 9M testis compared with 3M and 6M. Therefore, to ensure that sheep reach sexual maturity, we chose 9M as the sexual maturity stage.

[1] Comprehensive analysis of long noncoding RNA and mRNA expression patterns in sheep testicular maturation. Biology of Reproduction, 2018

Point 3: If the authors identified 9 month old sheep with high/low fertility or 9 month old sheep, some having reached sexual maturity and others still lacking, the study would be more complete. The authors conclusion suffer from confirmation bias. They identified pathways associated with growth and reproduction and then fit the DE genes to those pathways without mention of other pathways showing enrichment of DE genes. The Discussion section primarily restates the results of the study with little to no interpretation of the data.

Response 3: Thanks for your suggestion, we have added additional results on top GO and KEGG functional enrichment analysis of DE genes in line 273-292 (Figure 6 and 7), and have rewrote the discussion section in line 368-392.

Point 4: As such, the manuscript comes across as a "data" paper without offering any mechanism.

Response 4: Thanks for your comments. This study aims to explore the DE lncRNA and mRNA between immature and mature sheep pituitary, and also investigate the potential roles of lncRNA in pituitary. We not only analyzed the growth and reproduction related hormone levels, but also performed bioinformatics analysis by high-throughput sequencing and functional enrichment analysis. Moreover, several candidate lncRNAs related with pituitary development were screened, as well as the targeting relationship of lncRNA TCONS_00066406 and HSD17B12 were validated in vitro sheep pituitary cells using RNA interference and cell transfection.

Minor Revisions:

Point 1: There must be editing and correction of language and word usage. Soundness of experiments and scientific rigor is difficult to determine sometimes as the vocabulary and language has been, perhaps, misinterpreted. Section 2.6 in Materials and Methods: The authors utilize criteria such as p<0.05 and fold change>2 but neglect to include the false discovery rate as is the more accepted metric when using EdgeR for DE gene analysis.

Response 1: Thanks for your comments and suggestions. We have revised the language carefully. In our study, we used false discovery rate (FDR) as the criteria for differential expressed genes (DEGs) filter, however, the DEGs results were not expected. Thus, referring to other studies as follows [1], we utilized p<0.05 and |fold change| >2 as the screening criterion of DEGs.

[1] Rapid and dynamic alternative splicing impacts the Arabidopsis cold response transcriptome. Plant cell, 2018, IF=8.228.

Point 2: The p-value may or may not be relevant but using FDR controls for multiple testing error. It doesn't appear the cDNA libraries were constructed in a strand-specific manner however the authors make assumptions about strandedness of the transcripts and use this to identify antisense lncRNA. These assumptions are likely based solely on canonical splicing but the authors do not state this.

Response 2: Thanks for your comments. We have not clearly elaborated the method of constructing cDNA libraries, and we have rewritten this part as follows “The total RNA of six pituitary samples were extracted using TRIzol (Invitrogen Life Technologies, Carlsbad, CA, USA) and the RNA purity and quality were detected by NanoDrop (NanoDrop Technologies, Wilmington, DE, USA). Then ribosomal RNAs (rRNAs) were removed to retain mRNAs and ncRNAs. The remain RNAs were randomly fragmentated into short fragments which were used for cDNA synthesis. And first-strand cDNA was transcribed with random hexamers primers. dNTPs, RNase H and DNA polymerase I were used for second-strand synthesis. Next, cDNA fragments purification, end repair, poly(A) addition and Illumina sequencing adapters ligation were carried out with QiaQuick PCR extraction kit. Then UNG (Uracil-N-Glycosylase) was used to digest the second-strand cDNA. cDNA fragments were size selected by agarose gel electrophoresis, PCR amplified, and sequenced using Illumina HiSeqTM 4000 by Gene Denovo Biotechnology Co. (Guangzhou, China) ” (in line 115-127).

Point 3: Further, current assemblies and gene predictions result in many possible non-canonical splice sites. I cannot determine if the sequencing data has been submitted to public repositories or not. Sections 3.3: Authors claim to identify 1,256 new lncRNA and 8,575 new mRNA. This seems extraordinary.

Response 3: Thanks for your comments. In our study, the reads were mapped to the genome Oar_ v1.0 that referenced from the NCBI database (https://www.ncbi.nlm.nih.gov/genome/?term=sheep) and we identified 9,831 new transcripts using Cufflinks software. Then, CPC and CNCI software were used to predict the coding ability of new transcripts, and the new transcripts were compared to the protein database SwissProt. The intersection of these transcripts without coding potential and protein annotation information was taken as reliable new lncRNA prediction results. Thus, 1256 new lncRNA were identified.

Point 4: The authors do not propose a reason for identifying so many novel transcripts. Are these simply new isoforms of currently annotated genes? Many of the novel lncRNA are intergenic. Are these relatively close to existing coding gene annotations? If so, it is entirely possible the novel transcripts are novel UTR exons in protein-coding genes that simply were not previously annotated.

Response 4: Thanks for your comments. In our study, firstly, we performed the transcripts reconstructing (identified new transcripts) by Cufflinks software, and then we reanalyzed the coding potential and protein annotation information of new transcripts. Finally, the new lncRNAs were filtered. In addition, we have added the method of new lncRNA identification in method section.

Reviewer 2 Report

In general, I think that the manuscript shows interesting results regarding the potential role of lncRNA involved in male immature and mature sheep pituitary glands and the genes potentially regulated by then which may influence hormone secretion. Moreover, in the study authors validate one of the lncRNAs, showing its effects over its target gene, LH and FSH.

However, there are major points that I strongly recommend to modify to increase the scientific soundness of the manuscript. 

First of all, I strongly recommend repeating the analyses with the last version of the Ovine reference genome, Oarv3.1.

Moreover, the software used for the alignment, TopHat2 is in low maintenance since 2016, and developers recommend to use HISAT2 which provides the same functionality in a more accurate and efficient way. Thus, I recommend do repeat the analyses with the new version of the genome and HISAT2 or other aligners for RNA-Seq that is actualized

Continuing with the material and methods, authors should indicate which is the threshold for the coding potential which is used to determine that a transcript is a lncRNA. 

Authors should indicate which were the filters for the quality control of the RNA-Seq raw data (fastq files)

It is also worthy to indicate which is the window size they used to determine that a lncRNA was cis-acting with a gene.

In the paragraph of the differential expression analyses, authors indicate that they use FPKM normalized counts and edgeR software. As far as I know, EdgeR performs its own normalization and requires raw counts. Could you explain?

Indicate the software used to perform the enrichment analyses. In addition,  could the authors clarify if the analyses performed were functional enrichment analyses or gene annotation? The terms are not clear in the text

Author Response

Response to Reviewer 2 Comments

In general, I think that the manuscript shows interesting results regarding the potential role of lncRNA involved in male immature and mature sheep pituitary glands and the genes potentially regulated by then which may influence hormone secretion. Moreover, in the study authors validate one of the lncRNAs, showing its effects over its target gene, LH and FSH.

However, there are major points that I strongly recommend to modify to increase the scientific soundness of the manuscript.

Point 1: First of all, I strongly recommend repeating the analyses with the last version of the Ovine reference genome, Oarv3.1.

Response 1: Thank you so much for your comments. In our study, we analyzed our sequencing data with Oar_ v1.0, and it is released to NCBI database in 2017. And Oarv3.1 was released in 2013. (https://www.ncbi.nlm.nih.gov/genome/genomes/83). Thus, we selected Oar_ v1.0 for analysis.

Point 2: Moreover, the software used for the alignment, TopHat2 is in low maintenance since 2016, and developers recommend to use HISAT2 which provides the same functionality in a more accurate and efficient way. Thus, I recommend do repeat the analyses with the new version of the genome and HISAT2 or other aligners for RNA-Seq that is actualized

Response 2: Thanks for your suggestions. We found that HISAT2 and TopHat2 were developed by the same research and development team. And HISAT2 is optimized in algorithm compared to TopHat2, but the difference is not significant. And TopHat2 software is used in many transcriptome articles [1-2]. Therefore, using TopHat2 software for transcriptome analysis is reasonable.

[1] Comparative Transcriptome Analysis of Gene Expression Patterns in Tomato Under Dynamic Light Conditions. Genes, 2019

[2] Systematic identification and analysis of heat-stress-responsive lncRNAs, circRNAs and miRNAs with associated co-expression and ceRNA networks in cucumber (Cucumis sativus L.). Physiologia plantarum,2019

Point 3: Continuing with the material and methods, authors should indicate which is the threshold for the coding potential which is used to determine that a transcript is a lncRNA.

Response 3: Thanks for your suggestions. We have added the threshold for coding potential (the CPC/CNCI score <0 is used to determine that a transcript is a lncRNA) in line 1141-142.

Point 4: Authors should indicate which were the filters for the quality control of the RNA-Seq raw data (fastq files)

Response 4: Thanks for your suggestions. We have added the filtering rules of raw data in method part (line 129-132).

Point 5: It is also worthy to indicate which is the window size they used to determine that a lncRNA was cis-acting with a gene.

Response 5: Thanks for your suggestions. We have added the window size “within 10kb” of cis-regulated interaction.

Point 6: In the paragraph of the differential expression analyses, authors indicate that they use FPKM normalized counts and edgeR software. As far as I know, EdgeR performs its own normalization and requires raw counts. Could you explain?

Response 6: Thank you so much for your comments. We have not described the method of differential expression analysis accurately, and we have rewritten this part as follows: The differentially expressed transcripts of mRNAs and lncRNAs were analyzed respectively. The differential expression analysis was performed using edgeR package (http://www.r-project.org/), and the p-value < 0.05 and fold change > 2 were the filtering criteria to identify significant differential expressed genes (DEGs). Then the differential expression pattern clustering was drawn by Gene Denovo platform (http://www.omicsmart.com/).

Point 7: Indicate the software used to perform the enrichment analyses. In addition,  could the authors clarify if the analyses performed were functional enrichment analyses or gene annotation? The terms are not clear in the text

Response 7: Thank you so much for your comments. We have not described the method of GO and KEGG enrichment analysis clearly, and we have rewritten this section in method part.